# Natural Molecules in the Management of Polycystic Ovary Syndrome (PCOS): An Analytical Review

**DOI:** 10.3390/nu13051677

**Published:** 2021-05-15

**Authors:** Matteo Iervolino, Elisa Lepore, Gianpiero Forte, Antonio Simone Laganà, Giovanni Buzzaccarini, Vittorio Unfer

**Affiliations:** 1R&D Department, Lo.Li. Pharma Srl, 00156 Rome, Italy; m.iervolino@lolipharma.it (M.I.); e.lepore@lolipharma.it (E.L.); g.forte@lolipharma.it (G.F.); 2Department of Obstetrics and Gynecology, “Filippo Del Ponte” Hospital, University of Insubria, 2100 Varese, Italy; antoniosimone.lagana@uninsubria.it; 3The Experts Group on Inositol in Basic and Clinical Research (EGOI), 00156 Rome, Italy; 4Unit of Gynecology and Obstetrics, Department of Women and Children’s Health, University of Padua, 35128 Padua, Italy; giovanni.buzzaccarini@gmail.com; 5Systems Biology Group Lab, Department of Experimental Medicine, Sapienza University of Rome, 00161 Rome, Italy

**Keywords:** polycystic ovary syndrome, myo-inositol, D-chiro-inositol, resveratrol, vitamin C, vitamin E, vitamin D, omega-3 fatty acids

## Abstract

Polycystic ovary syndrome (PCOS) is a heterogenous disorder characterized by chronic ovulation dysfunction and hyperandrogenism. It is considered the most common endocrinological disorder, affecting up to 25% of women of reproductive age, and associated with long-term metabolic abnormalities predisposing to cardiovascular risk, such as insulin resistance (IR), dyslipidemia, endothelial dysfunction, and systemic inflammation. PCOS is also characterized by elevated serum levels of luteinizing hormone (LH), causing a condition of hyperandrogenism and a consequent altered ratio between LH and the follicle stimulating hormone (FSH). Over the years, several different approaches have been proposed to alleviate PCOS symptoms. Supplementation with natural molecules such as inositols, resveratrol, flavonoids and flavones, vitamin C, vitamin E and vitamin D, and omega-3 fatty acids may contribute to overcoming PCOS pathological features, including the presence of immature oocyte, IR, hyperandrogenism, oxidative stress and inflammation. This review provides a comprehensive overview of the current knowledge about the efficacy of natural molecule supplementation in the management of PCOS.

## 1. Introduction

Polycystic ovary syndrome (PCOS) is considered the most common endocrinological disorder, affecting up to 25% of women throughout their reproductive ages [1,2]. It is often characterized by a condition of chronic oligo—or anovulation (usually manifested as oligo—or amenorrhea), and hyperandrogenism [3], which derives from elevated serum levels of luteinizing hormone (LH) and a consequent altered ratio between LH and follicle stimulating hormone (FSH) [4,5,6,7].

PCOS is a heterogenous disorder, also associated with long-term metabolic abnormalities such as insulin resistance (IR), dyslipidemia, endothelial dysfunction and systemic inflammation, predisposing patients to an earlier cardiovascular risk, compared to women unaffected by PCOS disorder.

The Rotterdam workshop consensus [8] established the diagnostic criteria for PCOS based on a combination of at least two of the following three clinical features: (1) chronic oligo-anovulation; (2) polycystic ovaries at ultrasound examination; (3) hyperandrogenism (clinical and/or biochemical), featuring acne, androgenic alopecia and hirsutism [9]. Accordingly, four different groups of PCOS patients were identified, those with:Chronic ovulatory disorder, hyperandrogenism and polycystic ovary;Chronic ovulatory disorder and hyperandrogenism;Hyperandrogenism and polycystic ovary;Chronic ovulatory disorder and polycystic ovary.

Across seven recent studies [10,11,12,13,14,15,16], the frequency of the first group, presenting all three PCOS diagnostic criteria, was between 52.8% and 71.0%. In comparison, the other groups exhibited highly variable frequencies among the different studies. Since these surveys were conducted in various parts of the world, statistics were certainly influenced by genetic, environmental, and cultural factors linked to lifestyle, which plays a crucial role in PCOS management.

Even though insulin resistance was not included in the Rotterdam criteria, it is well known that it is a recurring sign in PCOS women and deserves proper attention due to the associated potential cardiovascular risk. In fact, the key role of IR, and/or compensatory hyperinsulinemia in PCOS onset and progression, is largely supported by increasing evidence [17,18,19]. Regardless of weight, about 30–40% of lean PCOS patients, along with up to 80% of PCOS women with obesity of the upper body (increased waist circumference and waist-to-hip ratio), exhibit hyperinsulinemia secondary to IR [20,21]. In this context, obesity also exacerbates PCOS pathological features. Indeed, elevated levels of insulin induce an increase in free circulating androgen levels [22] and a consequent IR condition, which can enhance the risk of developing glucose intolerance, type 2 diabetes, and lipid abnormalities in PCOS women [23,24,25]. Numerous molecules participate in the insulin signaling pathway. Many of them come from natural sources and their concentration depends on daily intake via food. Thus, correct dietary habits help maintain physiological ovarian functions [26]. In the case of reduced intake due to a specific diet or to impaired absorption, supplementation with natural molecules such as inositols, resveratrol, flavonoids and flavanones, vitamin C, vitamin E, vitamin D and omega-3 fatty acids may contribute to overcoming PCOS related symptoms.

These are natural molecules representing different chemical compounds acting on various pathological aspects of PCOS, including ovarian functionality, hormonal and metabolic profile, inflammatory state and oxidative stress.

This review aims to provide an overview of the effects of supplementation with natural molecules in the management of PCOS, gathering the evidence from the most recent literature.

## 2. Inositol

Inositols were discovered in muscle tissue more than 150 years ago, but only in the last few decades they have attracted strong interest as precursors of inositol 3-phosphate (InsP_3_), which acts as a second messenger in several intracellular pathways. Inositols are chemical compounds referred to as carbocyclic polyols. They have the same brute formula of glucose (C_6_-H_12_-O_6_) and they are present in almost all forms of life.

Inositols naturally occur as five stereoisomers [27], with myo-inositol (Myo-Ins) and D-chiro-inositol (D-Chiro-Ins) the most abundant. They are involved in several biological processes (e.g., cytoskeleton assembly and intracellular calcium concentration control) and also in the endocrine modulation. Both isomers are second messengers of insulin, but they work through different mechanisms: Myo-Ins is involved in the expression of glucose transporters and in cellular glucose uptake, while D-Chiro-Ins is mainly involved in glycogen synthesis and storage. Notably, Myo-Ins is physiologically converted to D-Chiro-Ins by an insulin-dependent epimerase.

Inositol metabolism is impaired in women with PCOS. Specifically, PCOS patients are generally characterized by an altered ratio between Myo-Ins and D-Chiro-Ins, in favor of the former. In fact, PCOS women tend to exhibit insulin resistance, resulting in a reduced intracellular conversion of Myo-Ins to D-Chiro-Ins [28]. An opposite situation occurs in the ovaries, which maintain normal sensitivity to insulin [28], becoming enriched in D-Chiro-Ins and depleted in Myo-Ins. In such tissue, Myo-Ins acts as the second messenger of the FSH signaling pathway. Indeed, numerous studies reported that the dietary supplementation of Myo-Ins improves metabolic and hormonal parameters of PCOS women, positively affecting the menstrual cycle and oocyte quality [29,30].

A recent meta-analysis from Unfer et al. [29] evaluated the efficacy of inositol-based therapy, including nine randomized clinical trials (RCTs) with a total of 247 cases (PCOS women) and 249 controls (non-PCOS women). The selected studies investigated the effects of the supplementation of Myo-Ins, alone or in combination with D-Chiro-Ins, highlighting its beneficial effect in improving the metabolic profile of PCOS women, and in reducing their hyperandrogenism. Specifically, the treatment with Myo-Ins significantly decreased levels of insulin and androgens (free testosterone) and the HOMA-index, while it increased levels of sex hormone binding globulin (SHBG). In detail, studies reported that Myo-Ins significantly increased SHBG levels after at least 24 weeks of administration.

Another recent meta-analysis [30] further supports these findings, reporting that Myo-Ins supplementation significantly improves the rate of ovulation and regulates the frequency of menstrual cycles. Studies demonstrated that the supplementation of 2 g of Myo-Ins twice daily improved hormonal parameters in women affected by PCOS. Moreover, other evidence highlighted that Myo-Ins should be taken on an empty stomach to avoid absorption interference, and that administration should be repeated. Indeed, kinetic analyses reported a 12 h half-life of inositol [31], suggesting that a double administration may guarantee the correct plasma concentration throughout the day.

It is crucial to emphasize the safety of Myo-Ins administration. The U.S. Food and Drug Administration (FDA) included Myo-Ins in the list of compounds generally recognized as safe (GRAS), implying that it is considered safe by experts and that it meets the food additive tolerance requirements of the Federal Food, Drug, and Cosmetic Act (FFDCA).

The androgen excess often observed in insulin-resistant PCOS patients depends on compensatory hyperinsulinemia, which leads to an unbalanced ovarian Myo-Ins:D-Chiro-Ins ratio. While Myo-Ins maintains the physiological FSH ovarian signaling, D-Chiro-Ins contributes to reducing insulin resistance and systemic insulin levels. For this reason, an increasing number of studies have evaluated the combination of the two inositols as a treatment for PCOS patients with metabolic alterations. Among different ratios tested, several studies concluded that the average plasma ratio of 40:1 (Myo-Ins:D-Chiro-Ins) is the most effective approach to restore metabolic and endocrinological physiology in overweight or obese PCOS women (BMI > 25) [32].

The International Consensus Conference in Florence on the use of Myo-Ins and D-Chiro-Ins in obstetrics and gynecology stated that the 40:1 ratio has beneficial effects also in assisted reproductive technology (ART), improving oocyte and ovarian quality [33]. Furthermore, Colazingari and colleagues [34] demonstrated the advantages of using the 40:1 Myo-Ins:D-Chiro-Ins ratio for oocyte quality compared to D-Chiro-Ins supplementation alone.

In PCOS women, a positive correlation exists between the volume of follicular fluid, Myo-Ins levels, and the presence of mature oocytes. As the ovaries of these subjects are depleted in Myo-Ins [35], administration of high doses of D-Chiro-Ins understandably leads to decreased oocyte quality and ovarian response [36]. On the other hand, the administration of a combined treatment of Myo-Ins and D-Chiro-Ins provided the best results [37].

An interesting study on a PCOS mouse model revealed that treatment with a 40:1 Myo-Ins:D-Chiro-Ins ratio restored normal histological features and a proper thickness ratio of theca/granulosa cell layer (TGR), suggesting that the treatment efficiently reversed the androgenic phenotype [38].

As mentioned, insulin resistance and compensatory hyperinsulinemia are frequent dysfunctions in PCOS women, mainly associated with obesity, but present also in lean women. This metabolic alteration is an indicator for increased cardiovascular risk and for the development of other serious related diseases, including type 2 diabetes, hypertension, and metabolic syndrome [39]. In this regard, the therapy with Myo-Ins and D-Chiro-Ins in a 40:1 ratio may improve levels of low-density lipoproteins (LDL), high-density lipoprotein (HDL) and triglycerides (TG), at the same time reducing fasting and circulating insulin levels [40].

Interestingly, referring to inositol supplementation in PCOS patients, later studies highlighted that the addition of α-lactalbumin (α-LA) can optimize the beneficial effects in PCOS women, overcoming the common problem of inositol resistance occurring in these patients [41]. Montanino and colleagues indicated that following inositol treatment, only 62% of women ovulated, while 38% were resistant and did not ovulate. This resistant group was then treated with inositol plus α-LA, following which about 86% of them ovulated, accompanied by an improvement in hormone and lipid profile. Indeed, in vitro studies corroborated the ability of α-LA in improving the intestinal adsorption of inositols, ensuring a higher effectiveness of inositol-based therapy in PCOS condition [31].

In conclusion, treatment with the combination of Myo-Ins and D-Chiro-Ins in the 40:1 ratio seems to be the most effective approach for restoring ovulation and normalizing crucial parameters (progesterone, LH, SHBG, estradiol, and testosterone) in overweight and obese PCOS patients [42], also reducing the risk of cardiovascular-related problems.

Overall, available evidence indicates that the positive effects of the combined therapy of Myo-Ins and D-Chiro-Ins can be related to the regulation of glucose metabolism, which is guaranteed by the simultaneous administration of the two stereoisomers in the physiological average plasma ratio.

## 3. Resveratrol, Flavonoids and Flavanones

Resveratrol is a natural polyphenol found in grapes, nuts, and berries, with marked anti-inflammatory and antioxidant effects, and cardioprotective properties. Resveratrol was suggested as a potential therapeutic agent in the treatment of infertility, which is a condition related to diminished ovarian reserve, obesity, and PCOS [43,44,45,46,47].

However, most recent scientific evidence suggests that resveratrol should be avoided during the luteal phase and pregnancy, due to the anti-deciduogenic function in uterine endometrial tissue. Specifically, resveratrol seems to inhibit the expression of cellular retinoic acid-binding protein 2 (CRABP2-RAR), avoiding the decidualization process and decidual senescence. At the same time, it induces the deacetylation of crucial decidual genes [48,49,50,51] encoding for prolactin (PRL) and insulin-like growth factor-binding protein-1 (IGFBP1).

Moreover, the teratogenicity of resveratrol is still debated, and clinical studies have demonstrated a reduction in the rate of clinical pregnancies and a statistically significant increase in abortion rate, compared with age-matched controls, in the practice of ART [48,49,50,51].

Benrick and colleagues [52] investigated the effects of resveratrol on insulin resistance, studying the outcomes of 5/6-week treatment in PCOS rats. The authors demonstrated that supplementation failed to improve insulin sensitivity, while physical exercise restored physiological insulin sensitivity. Unlike physical exercise, resveratrol seems to have no beneficial effects on fat mass, adipocyte size, and estrus cyclicity.

More recent evidence suggests that resveratrol also has anti-inflammatory, antioxidative, anti-apoptotic properties. Indeed, several studies indicated that resveratrol seems to inhibit the expression of pro-inflammatory cytokines, such as interleukin-1β (IL-1β), interleukin-6 (IL-6), and cyclooxygenase-2 (COX-2), through the modulation of the NF-κB pathway [51] by inhibiting IκB kinase activity [53,54]. However, few clinical trials have investigated its effects on inflammatory pathways in PCOS women; therefore, other studies are necessary [54,55,56].

Flavonoids and flavanones constitute a large group of plant secondary metabolites with pharmacological potential [57]. Naringenin is a natural flavanone derived from grapefruit and various plant species [58,59]. Several recent studies highlighted its beneficial role in PCOS mouse models with cytoprotective and anti-inflammatory effects [60]. Naringenin can reduce testosterone and estradiol levels in PCOS women, and it can increase the concentration of enzymes involved in the scavenging of reactive oxygen species (ROS) [61]. Indeed, oxidative stress is indicated as one of the pathogenetic features of PCOS; therefore, molecules that can scavenge ROS may be beneficial in PCOS treatment. Furthermore, Hong and colleagues demonstrated that naringenin can prevent weight gain associated with PCOS and it can cause a reduction in the serum glucose levels of PCOS rats [62].

Interestingly, naringenin is not the only flavonoid found in a variety of plants. Namely, rutin is a citrus flavonoid glycoside exhibiting positive effects in the treatment of PCOS. A recent study revealed that rutin can ameliorate obesity and insulin resistance in obese mice, by enhancing the activity of the brown adipose tissue (BAT) and inducing the formation of the beige adipocytes in white adipose tissue (WAT) [63]. In line with this, Hu and colleagues demonstrated that rutin treatment significantly activates the BAT-ameliorating PCOS phenotype including hyperandrogenism, cyclicity and infertility [64].

## 4. Vitamin C

Vitamin C (or ascorbic acid) is a micronutrient, a molecule required by the body in small quantity, necessary for the physiological and healthy growth of cells and tissues. Being water-soluble, vitamin C is easily excreted in urine, and constant dietary intake is necessary.

Vitamin C exhibits antioxidant activities as it scavenges peroxyl radicals and restores the antioxidant properties of fat-soluble vitamin E. The overall outcome is the beneficial control of lipid peroxidation of intracellular and plasma membranes, similar to the antioxidant effect.

Olaniyan and colleagues [65] investigated the ovarian metabolic changes in PCOS Wistar rats, associated with vitamin C administration. They observed that vitamin C plays an important role in the regulation of the menstrual cycle and ovarian function, and that its levels are modulated throughout the menstrual cycle. Vitamin C levels decline immediately before ovulation, and increase again after post-ovulation temperature rise. This evidence is in line with the uptake of ascorbic acid in the pre-ovulatory phase, likely to facilitate proper ovulation. Ascorbic acid stimulates progesterone and oxytocin production, and high concentrations are present in the corpus luteum [66]. Moreover, ascorbic acid in the ovaries may be responsible for collagen synthesis, necessary for follicle and corpus luteum growth, as well as for post-ovulation repair of the ovarian tissue. Notably, impairment of these functions may contribute to the development of ovarian cysts [67].

On these premises, vitamin C deserves to be investigated as a potential therapeutic agent to improve ovarian morphology and anovulation associated with PCOS [65]. It is noteworthy that very few clinical trials have been conducted so far, and further studies are necessary to evaluate vitamin C effectiveness in the treatment of PCOS.

## 5. Vitamin E

Vitamin E (or tocopherol) is a fat-soluble vitamin that can be stored in the liver and released in small quantities to maintain physiological levels. Vitamin E exhibits antioxidant properties as it neutralizes free radicals and promotes cell renewal [68].

Recent evidence has confirmed that vitamin E may improve endometrial thickness in women with idiopathic infertility, thanks to its anticoagulant and antioxidant properties [69]. In addition, cotreatment with vitamin E and coenzyme Q10 for 8 weeks increased circulating levels of SHBG in PCOS patients [70], reducing free plasma testosterone concentrations.

Recently, Chen and colleagues [71] explored whether short-term supplementation with vitamin E leads to improved reproductive performance in the induction of ovulation in PCOS women, and whether associations between vitamin E and pregnancy rates exist. They observed that the treatment reduces oxidative stress, consequently reducing the exogenous human menopausal gonadotropin (HMG) dosage, with economic benefits for medical practice [71]. However, the supplementation of vitamin E seems to have a negligible effect on pregnancy rate [71].

## 6. Vitamin D

Vitamin D is a secosteroid hormone with progesterone-like activity [72], well known for maintaining calcium homeostasis and promoting bone mineralization [73]. The name of vitamin D usually refers only to when this molecule is administered exogenously in cases of hormone D deficiency [74].

To become active, vitamin D requires two hydroxylation steps, yielding first calcidiol (or calcifediol or 25(OH)D), and eventually calcitriol (1,25(OH)2D). Calcidiol and calcitriol are considered a pre-hormone and hormone, respectively. A growing body of literature suggests mechanistic implications of hormone D deficiency in insulin resistance, inflammation, dyslipidemia and decreased fertility, namely, clinical and metabolic phenomena commonly encountered in PCOS [75,76,77,78].

Kadoura S. and colleagues investigated the effects of combining calcium and vitamin D supplements with metformin on menstrual cycle abnormalities, gonadotropins and the IGF-1 system in vitamin D-deficient/insufficient PCOS women. In this randomized, placebo-controlled clinical trial, 40 PCOS women with low 25-OH-vitamin D serum levels (<30 ng/mL), were randomly assigned to take either metformin (1500 mg/daily) plus a placebo, or metformin (1500 mg/daily) plus calcium (1000 mg/daily) and vitamin D3 (6000 IU/daily) orally for 8 weeks. The authors observed that calcium and vitamin D supplements can support metformin effect on the regulation of menstrual cycle irregularity in vitamin D-deficient/insufficient PCOS patients, but this effect is not associated with any significant changes in gonadotropins or the IGF-1 system [79].

The meta-analysis by Miao and co-workers, concerning 11 studies, aimed to evaluate the effect of vitamin D supplementation on 483 women with PCOS. The main outcomes included body mass index (BMI), total testosterone, dehydroepiandrosterone sulfate (DHEAs), homeostasis model assessment of insulin resistance (HOMA-IR), homeostasis model assessment of β-cell function (HOMA-B), triglycerides, total cholesterol, or low-density lipoprotein-cholesterol. The results failed to show a positive effect of vitamin D supplementation on BMI, dehydroepiandrosterone sulphate, triglyceride levels or high-density lipoprotein-cholesterol. The data from the available randomized controlled trials (RCTs) on vitamin D indicate that supplementation may reduce insulin resistance and hyperandrogenism in patients with PCOS [80]. However, the evidence reported seems to still be contradictory, and further studies are necessary to validate the findings and draw conclusive results.

In a double-blind, randomized, placebo-controlled trial, Trummer and colleagues randomized 180 PCOS women to receive either vitamin D (20,000 IU/week) or a placebo for 24 weeks [81]. Supplementation with vitamin D led to a decrease in plasma glucose, one hour after the oral glucose tolerance test (OGTT), compared to the placebo.

Furthermore, in 2014, Lerchbaum E. et al. investigated the role of vitamin D in modulating the human reproductive process. They observed a greater thickness of the endometrium in women who had normal levels of vitamin D, which resulted in a better chance of becoming pregnant [82].

The molecular mechanism between vitamin D supplementation and improvement of PCOS is currently unknown. However, recent studies have claimed there are positive effects of vitamin D3 replacement in the treatment of PCOS. Indeed, vitamin D supplementation may attenuate the harmful effects of advanced glycation end products (AGEs) in women with PCOS, by enhancing androgen synthesis and improving abnormal folliculogenesis [83,84]. Namely, vitamin D attenuates the adverse effects of AGEs on steroidogenesis by human granulosa cells (GCs), possibly by downregulating the expression of the pro-inflammatory cell membrane receptor for AGEs (RAGE) [83,84].

All these results highlighted the potential positive effects of vitamin D supplementation on different pathologic features of PCOS. Nevertheless, more studies are needed to ascertain the benefits of vitamin D supplementation in PCOS management, also in association with other molecules.

## 7. Omega-3 Fatty Acids

Omega-3 belong to the class of polyunsaturated fatty acids (PUFAs). Among them, eicosapentaenoic acid (EPA) and docosahexaenoic acid (DHA) are the most biologically active, and they are mainly present in fatty fish such as salmon, mackerel, tuna, herring, and other types of small and blue fish.

Omega-3 fatty acids have antioxidant, anti-inflammatory, anti-obesity, and insulin-sensitizing properties [85,86,87]. In detail, they can improve insulin sensitivity by decreasing the production of inflammatory cytokines, including tumor necrosis factor-α (TNFα) and IL-6, and by increasing the secretion of anti-inflammatory adiponectin [88]. The positive effect of EPA and DHA supplementation on inflammatory processes and cardiovascular parameters have long been studied [89,90] in pathological contexts such as obesity [91], atherosclerosis [92], and diabetes mellitus [93]. Of note, PCOS disorder commonly involves insulin resistance and obesity, predisposing to cardiometabolic-related alterations (dyslipidemia, diabetes, hypertension), which usually occur after 40 years of age in affected women [94,95]. Interestingly, several studies investigated the effect of omega-3 fatty acids administration in PCOS women, with inconclusive results. While Sadeghi and colleagues [96] observed that omega-3 supplementation may have no beneficial effects on insulin resistance in PCOS patients, Khani B. [97] and co-workers reported that a 6-month treatment with omega-3 fatty acids improves waist circumference, HDL, LDL, triglycerides, and regularity of menstrual cycle compared to non-PCOS subjects. However, no significant changes were observed in other parameters such as body weight, number of ovarian follicles, size of ovaries, bleeding volume, menstrual bleeding, and hirsutism score, between intervention and control groups. On the other hand, a meta-analysis by Yang K. and colleagues [98] reported that omega-3 fatty acids positively affect insulin resistance (improving HOMA index and increasing adiponectin levels) and decrease the level of total cholesterol (TC), TG and LDL. However, there are no indications that omega-3 fatty acids directly affect BMI, fasting insulin, fasting glucose, and levels of HDL, FSH, LH, SHGB and total testosterone [98]. These results further indicate that omega-3 fatty acids increase insulin sensitivity, also acting on inflammatory state, stimulating the production of anti-inflammatory adipokine and reducing proinflammatory cytokines. However, the above mentioned analysis of Yang has some limitations, including small sample sizes and the short duration of omega-3 administration. Indeed, beneficial effects derived from treatments with omega-3 fatty acids over 6 months are still largely under-investigated.

Recently, Tosatti and colleagues [99] studied the influence of omega-3 fatty acid supplementation on inflammatory and oxidative stress markers in PCOS patients, through a systematic literature search of Medline/PubMed, Cochrane Central Register of Controlled Trials, Scopus, and Lilacs, until November 2019. They retrieved data from 323 studies suggesting that supplementation with omega-3 fatty acids could reduce the inflammatory state in PCOS women, due to decreased high-sensitivity C-reactive protein (hs-CRP) and increased adiponectin levels [99].

Based on evidence regarding the positive effects of omega-3 supplementation on inflammatory state and cardiometabolic alterations, and considering that cardiovascular problems may occur only after 40 years of age in PCOS women, a generalized administration of omega-3 fatty acids in all PCOS patients, regardless of age and needs, seems unnecessary. Therefore, administration of omega-3 fatty acids should be specific only for PCOS women who already exhibit inflammatory and cardiovascular-related symptoms, which usually occur between 40 and 45 years of age [94,95]. Daily intake of omega-3 fatty acids should range between 0.5 and 2% of energy requirement at any age, according to the latest revision of LARNs (the Reference Intake Levels of nutrients and energy for the Italian population). Moreover, studies on cardiovascular and anti-inflammatory effects in adults reported that the recommended minimum dose for combined EPA:DHA administration is 500 mg/day, reaching 2000–4000 mg/day in patients with recent myocardial infarction or in those with altered triglycerides levels [100].

Possible occurrence of side effects related to omega-3 fatty acids supplementation should be considered. They include mild gastrointestinal discomfort, intestinal gas (especially if the source of omega-3 fatty acids is fish oil), nausea, diarrhea and headache [101]. Notably, the intake of omega-3 fatty acids is contraindicated during antiplatelet and anticoagulant treatment, because they exert synergistic effects [102]. Consulting with doctors is always advised before beginning supplementation with omega-3 fatty acids, in order to rule out possible side effects or interference with other drugs, especially during pregnancy or breastfeeding. Indeed, current guidelines recommend avoiding EPA administration during pregnancy because of the possible competitive effects with arachidonic acid, which is essential for growth processes at the fetal stage [103]. Finally, omega-3 fatty acids have a higher calorific value compared to other dietary supplements, and their use should be carefully evaluated in obese or overweight patients to avoid negative impact on metabolic alterations, which are common in PCOS women.

## 8. Conclusions

Natural molecules reported here represent different chemical compounds acting with several mechanisms of action on pathological aspects of PCOS, such as ovarian functionality, hormonal and metabolic profile, inflammatory state, and oxidative stress (Appendix A).

The administration of Myo-Ins and D-Chiro-Ins in the 40:1 ratio seems to be the most effective choice to restore ovulation in PCOS women and to normalize hormonal parameters (progesterone, LH, SHBG, estradiol, and testosterone). Such an approach proved to consistently ameliorate the metabolic profile of obese PCOS patients, reducing the risk of cardiovascular problems. Beside inositols, other natural antioxidant and anti-inflammatory molecules seem to be effective in the management of PCOS. These natural molecules include resveratrol, flavonoids and flavanones, such as naringenin and rutin, vitamins such as vitamin C, vitamin E and vitamin D, and omega-3 fatty acids. In particular, omega-3 fatty acids appeared to improve symptoms of PCOS in women over 40 years of age. However, careful monitoring is necessary because of both possible adverse effects and interference with some pharmacological treatments. Finally, guidelines on the correct use of dietary EPA and DHA supplementation in pregnancy will discourage the administration of EPA.

## Data Availability

This is not applicable, since this review does not present any innovative data.

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
