# Peer review of "Natural Molecules in the Management of Polycystic Ovary Syndrome (PCOS): An Analytical Review"

_nutrients, 2021, doi:10.3390/nu13051677_

Round 1

Reviewer 1 Report

Comments and Suggestions for Authors

Although PCOS and its treatment are widely described in the literature, many aspects of this disorder are still under investigation. The publication by Matteo Iervolino et al. is well written and presents in a concise and understandable way the latest reports on natural compounds used in the treatment of PCOS. 

However I have minor comments that should help improve the paper:

  1. The introduction should contain information about the selection criteria for the described compounds.Please consider whether these natural molecules are representatives of different chemical groups or compounds with different properties or activities?

  1. The authors prove the effect of vitamin E, but in the cited publication (61) vitamin E itself did not significantly affect the tested parameters, only the coenzyme q10. Therefore, in addition to the effect of vitamin E, I suggest discussing the effect of vitamin D and its role in the pathogenesis of PCOS and PCOS-related complications. Authors can create a separate paragraph or include selected information in the chapter describing the effects of vitamin E. Many interesting papers are focused on this topic, which I suggest to read and cite, for example:
  • Effect of Calcium and Vitamin D Supplements as an Adjuvant Therapy to Metformin on Menstrual Cycle Abnormalities, Hormonal Profile, and IGF-1 System in Polycystic Ovary Syndrome Patients: A Randomized, Placebo-Controlled Clinical Trial. Sally Kadoura, Marwan Alhalabi, Abdul Hakim Nattouf; Adv Pharmacol Sci. 2019; 2019: 9680390.
  • Effect of vitamin D supplementation on polycystic ovary syndrome: A meta-analysis. Chen-Yun Miao, Xiao-Jie Fang, Yun Chen, Qin Zhang; Exp Ther Med. 2020 Apr; 19(4): 2641–2649.
  • Effects of vitamin D supplementation on metabolic and endocrine parameters in PCOS: a randomized-controlled trial. Christian Trummer, Verena Schwetz, Martina Kollmann, Monika Wölfler, Julia Münzker, Thomas R. Pieber, Stefan Pilz, Annemieke C. Heijboer, Barbara Obermayer-Pietsch, Elisabeth Lerchbaum; Eur J Nutr. 2019; 58(5): 2019–2028.

  1. Describing the main mechanism of action of the supplements mentioned in the publication (or groups of supplements) in relation to individual symptoms of PCOS would increase the scientific value of the publication. I suggest a schema form that will enrich the text.

  1. Please pay attention to the unification the References. Below some examples:

  • most items contain abbreviations of journal names, e.g. 1, 2, 3, 4, 5 while some contain full journal names, e.g. 63, 74, 75;
  • the journal abbreviations contain dots, e.g. 8, 32, 33, 34 or not;
  • after the year of publication the month is abbreviated, e.g. item 1, 2, 7, 10, 11 or not, e.g. 3, 4, 5, 6, 8;
  • similarly, some items contain numbers of issues in parentheses after volumes or whole items differ from the others, e.g. 48, 57, etc.
